# Latticed Gold Nanoparticle Conjugation via Monomeric Streptavidin in Lateral Flow Assay for Detection of Autoantibody to Interferon-Gamma

**DOI:** 10.3390/diagnostics11060987

**Published:** 2021-05-29

**Authors:** Weeraya Thongkum, Umpa Yasamut, Koollawat Chupradit, Supachai Sakkhachornphop, Jiraprapa Wipasa, Kanokporn Sornsuwan, On-anong Juntit, Rawiwan Pornprasit, Wanwisa Thongkamwitoon, Jirapan Chaichanan, Jaruwan Khaoplab, Chonnikarn Chanpradab, Watchara Kasinrerk, Chatchai Tayapiwatana

**Affiliations:** 1Division of Clinical Immunology, Department of Medical Technology, Faculty of Associated Medical Sciences, Chiang Mai University, Chiang Mai 50200, Thailand; weeraya.t@cmu.ac.th (W.T.); umpa.yas@cmu.ac.th (U.Y.); koollawat_c@cmu.ac.th (K.C.); kanokporn_sornsuwan@cmu.ac.th (K.S.); Onanong_jun@cmu.ac.th (O.-a.J.); watchara_kasinrerk@hotmail.com (W.K.); 2Center of Biomolecular Therapy and Diagnostic, Faculty of Associated Medical Sciences, Chiang Mai University, Chiang Mai 50200, Thailand; ssakkhachornphop@yahoo.com; 3Center of Innovative Immunodiagnostic Development, Department of Medical Technology, Faculty of Associated Medical Sciences, Chiang Mai University, Chiang Mai 50200, Thailand; 4Research Institute for Health Sciences, Chiang Mai University, Chiang Mai 50200, Thailand; jwipasa@hotmail.com; 5Bio Innovation Building, Mahidol University, Nakhon Pathom 73170, Thailand; rawiwan.por@mahidol.ac.th (R.P.); w.chaisaklert@gmail.com (W.T.); jirapan.cha@mahidol.ac.th (J.C.); jaruwan.kho@mahidol.ac.th (J.K.); chonnikarn.c@ku.th (C.C.)

**Keywords:** interferon-γ, anti-interferon-γ autoantibody, conventional indirect ELISA, immunochromatographic strip test

## Abstract

Adult-onset immunodeficiency syndrome (AOID) patients with autoantibodies (autoAbs) against interferon-gamma (IFN-γ) generally suffer from recurrent and recalcitrant disseminated non-tuberculous mycobacterial diseases. Since the early stages of AOID do not present specific symptoms, diagnosis and treatment of the condition are not practical. A simplified diagnostic method for differentiating AOID from other immunodeficiencies, such as HIV infection, was created. Anti-IFN-γ is generally identified using enzyme-linked immunosorbent assay (ELISA), which involves an instrument and a cumbersome process. Recombinant IFN-γ indirectly conjugated to colloidal gold was used in the modified immunochromatographic (IC) strips. The biotinylated-IFN-γ was incorporated with colloidal-gold-labeled 6HIS-maltose binding protein-monomeric streptavidin (^6HIS^MBP-mSA) and absorbed at the conjugate pad. The efficacy of the IC strip upon applying an anti-IFN-γ autoAb cut-off ELISA titer of 2500, the sensitivity and specificity were 84% and 90.24%, respectively. When a cut-off ELISA titer of 500 was applied, the sensitivity and specificity were 73.52% and 100%, respectively.

## 1. Introduction

Interferon-gamma (IFN-γ) is secreted by T-cells and natural killer cells. IFN-γ is one of the vital cytokines associated with anti-microbial responses, antigen processing, inflammation, macrophage differentiation, growth inhibition, cell death, tumor immunity, and autoimmunity [1]. IFN-γ signaling is crucial in host defense mechanisms against intracellular pathogens.

The interleukin 12 (IL-12)/IFN-γ axis of the immune system plays a role against mycobacteria and viruses. Mendelian susceptibility to mycobacterial diseases and monocytopenia and mycobacterial infection syndrome are classified as early- and late-onset immunodeficiencies that occur upon defects in the genes involving the IL-12/IFN-γ pathway [2,3]. Besides these genetic defects, autoimmunity with anti-IFN-γ autoAbs is associated with non-tuberculous mycobacterial (NTM) infections. Since adults comprise the major population that suffers from this autoimmune disease, it is called adult-onset immunodeficiency (AOID). The prevalence of AOID is increasing over the years, especially in Asian countries [4,5,6]. Some reports have proposed that genetic factors and cell-mediated immunity may trigger the disease [7,8,9]. Owing to the immunological functions of IFN-γ, patients with anti-IFN-γ autoAbs suffer from life-threatening opportunistic infections. The handling of disseminated opportunistic infections in patients who produce anti-IFN-γ autoAbs is difficult. Patients with persistent, progressive, or severe anti-IFN-γ autoAb-related infections require standard antimicrobial therapy and adjunctive therapy such as IFN-γ, immunoglobulin, and plasmapheresis [10,11,12,13]. However, the success of the therapy is limited in patients with IFN-γ autoAb syndrome, since the anti-IFN-γ autoAbs neutralize IFN-γ post its infusion. Consequently, the detection of anti-IFN-γ autoAbs will improve the treatment regimens in these patients.

In a clinical laboratory, anti-IFN-γ Abs and neutralizing activity have been investigated to verify immunodeficiency caused by autoAbs. The levels of total anti-IFN-γ autoAbs and neutralizing activity have been assessed using indirect enzyme-linked immunosorbent assay (ELISA) and IFN-γ-mediated Signal transducer and activator of transcription 1 (STAT1) phosphorylation, respectively [5,14,15]. However, the same is not practical for general diagnostic laboratories because the latter requires flow cytometry. Recently, QuantiFERON-TB Gold In-tube (QFT-GIT), a commercialized IFN-γ release assay, has been used to screen neutralizing anti-IFN-γ autoAbs in patient sera. An indeterminate QFT-GIT result in the mitogen tube implies the presence of neutralizing antibodies [16]. However, it cannot be clearly interpreted that an indeterminate QFT-GIT result is caused by the presence of neutralizing anti-IFN-γ autoAbs. Recently, a dot ELISA has been developed by Rattanathammethee et al., to detect anti-IFN-γ Abs in AOID patient samples [17]. Another method, such as a biosensor, has been developed to verify the interaction of an anti-IFN-γ Ab with IFN-γ protein [18]. However, these methods are technically demanding, time-consuming, and require complex expensive equipment, limiting their availability in resource-limited settings. Thus, a simplified assay must be developed to support the convenience of AOID diagnosis.

The antibody- or protein-labeling technique is commonly used as a tracer element in the immunochromatographic (IC) strip test, for interpretation of results using the naked eye. The avidin or streptavidin-biotin system preferentially enhances the signal of the immunoassay [19,20]. However, the production of avidin and streptavidin is not cost-effective. Recently, a novel monomeric streptavidin (mSA) has been generated. mSA is an engineered monomeric streptavidin protein. mSA is advantageous since its active form can be simply produced in the standard *Escherichia coli* (*E. coli*) expression system [21]. Various applications of mSA have been reported based on its unique ability to bind biotin molecules. Recombinant mSA has been formerly applied in flow cytometry analysis and ELISA assay [22]. Colloidal gold nanoparticles have been used for decades due to the properties of colloidal gold such as surface chemistry, shape, size-related electronic, and optical properties [23,24]. The ability to conjugate proteins to colloidal gold affords a wide variety of probes for biosensor and lateral flow assay [19,25,26]. In this study, purified ^6HIS^MBP-mSA protein was conjugated with colloidal gold and tangled with biotinylated-IFN-γ. This complex served as a detection system for the IC strip. The modified IC strip test was validated by comparing its efficiency with that of conventional ELISA. The obtained data will provide an alternative assay for simplifying the detection of anti-IFN-γ autoAbs.

## 2. Materials and Methods

### 2.1. Study Population

The serum samples used in this study were obtained from the Research Institute for Health Sciences, Chiang Mai University, Thailand. The samples were classified into three groups: 18 active AOID cases, 18 inactive AOID cases, and 30 normal healthy subjects. The patients (male or female) were enrolled in this study with clinical identification consisting of at least one episode of NTM or correlated opportunistic infections, in addition to being HIV-negative and anti-IFN-γ antibody-positive, according to the criteria of AOID cases. Active AOID patients had symptoms and signs of infections, as described in the criteria, while inactive AOID patients were AOID cases with no symptoms or signs of infection. Healthy individual controls (male or female) were HIV- and anti-IFN-γ antibody-negative and had no infections or immunosuppressive status.

### 2.2. Ethical Approval

The study was approved by the ethics committees of the Faculty of Medicine (project number 105/2557, 13 March 2014) and the Research Institute for Health Sciences of Chiang Mai University (project number 13/56, 18 April 2014). An open discussion on details of the projects including procedures involved was conducted with potential participants, and those who willing to participate were enrolled with written consent.

### 2.3. Type of Cells

*Escherichia coli* strain BL21 (DE3) purchased from Sigma-Aldrich (Merck, Germany) was used as a host for recombinant protein production.

### 2.4. Production of IFN-γ Protein

An isolated colony of *E. coli* BL21 (DE3) containing a plasmid coding for histidine-tagged human IFN-γ gene with ampicillin-resistant gene was pre-cultured in 10 mL of Super Broth (SB) medium with 100 µg/mL ampicillin. The cultures were grown overnight at 37 °C, with shaking. The overnight culture was transferred into the same medium in a 500 mL culture flask, and the bacteria were cultured until the optical density (OD) at 600 nm reached a value of 0.5. The culture was then transferred to a 5 L fermenter containing SB medium. Fermentation with the optimized fed-batch parameters was carried out at 37 °C, with aeration and agitation, until the OD of the culture was 0.8 at 600 nm. To induce protein expression, 1 mM isopropyl β-D-1-thiogalactopyranoside (IPTG) (Thermo Fisher Scientific, Waltham, MA, USA) was added to the medium. Cultures were grown for an additional 18 h at 30 °C. The cells were then harvested by centrifugation at 3000 rpm for 30 min at 4 °C. The protein was purified using a HisTrap™ HP column (GE Healthcare, Uppsala, Sweden). Protein purity was validated using sodium dodecyl sulfate-polyacrylamide gel electrophoresis (SDS-PAGE) with PAGE Blue™ Protein Staining Solution (Thermo Scientific, Rockford, IL, USA). The IFN-γ protein was examined using western blot with a mouse anti-His6-tagged antibody (ABM Inc., Richmond, BC, Canada) and anti-IFN-γ monoclonal antibody (anti-IFN-γ mAb; clone B27; ImmunoTools, Friesoythe, Germany). The membranes were then washed with washing buffer (0.05% Tween-20 in phosphate-buffered saline (PBS), pH 7.4) five times, followed by incubation with goat anti-mouse immunoglobulins (goat anti-mouse Igs) conjugated with horseradish peroxidase (HRP) (KPL, Gaithersburg, MD, USA) at room temperature for 1 h. After the washing step, the signal was generated using a 3,3,5,5-tetramethylbenzidine (TMB) Membrane Peroxidase Substrate System (KPL, Gaithersburg, MD, USA).

### 2.5. Expression of ^6HIS^MBP-mSA

The expression of ^6HIS^MBP-mSA in the bacterial system has been reported [21]. A modified method of the same was used in this study. The pET -MBP-mSA2 plasmid [a gift from Dr. Sheldon Park, plasmid number 52319, (Addgene, Watertown, MA, USA)] carrying mSA gene, 6-histidine tag gene, and maltose-binding protein-monomeric streptavidin (^6HIS^MBP-mSA) was transformed into *E. coli* BL21 (DE3). Then, a single colony of transformed *E. coli* was picked and cultured in SB medium with 50 µg/mL kanamycin antibiotic overnight at 37 °C, with shaking at 200 rpm. Ten mL of the overnight culture was then transferred to 90 mL of SB medium [with 50 µg/mL of kanamycin antibiotic and 0.05% (*w*/*v*) of glucose] and incubated with shaking at 37 °C until the OD reached a value of 0.8 at 600 nm. Expression of ^6HIS^MBP-mSA was induced by adding 0.05 mM IPTG, followed by incubation with shaking at 20 °C for 16 h. The cells were then harvested by carrying out a round of centrifugation at 3000 rpm for 30 min at 4 °C. The protein was purified using a HisTrap^TM^ HP column. The purified proteins were verified using SDS-PAGE and western blot. The membranes were incubated with an anti-His_6_-tagged antibody and washed five times with the washing buffer. Then, the membrane was incubated with goat anti-mouse Igs conjugated with HRP at room temperature for 1 h. The signal was developed using the TMB Membrane Peroxidase Substrate System.

### 2.6. Detection of Anti-IFN-γ AutoAbs Using Conventional Indirect ELISA Assay

In this study, a conventional indirect ELISA was used to detect anti-IFN-γ autoAbs in the sera. Initially, microtiter plates were coated with 50 μL of 10 µg/mL purified recombinant IFN-γ diluted in coating buffer (1 M NaHCO_3_, pH 9.6) by overnight incubation of the coated plate at 4 °C under moist conditions. The coated wells were washed three times with washing buffer and blocked with 2% (*w*/*v*) skimmed milk in PBS (pH 7.4) for 1 h at room temperature. Serum samples were diluted in the ratios of 1:500, 1:2500, and 1:12,500 in PBS with 2% skimmed milk, followed by the addition of 50 µL of each diluted serum sample to the wells and incubation at room temperature for 1 h. The wells were then washed and incubated with rabbit anti-human IgG conjugated with HRP (KPL). After incubation, the wells were washed, and 50 µL of TMB substrate was added to them. The reaction was subsequently stopped, followed by measurement of the OD at 450 nm using an ELISA reader (Hercuvan Lab Systems, Milton Cambridge, UK).

### 2.7. Biotinylation of Recombinant IFN-γ

Recombinant IFN-γ protein was immobilized with biotin molecules using an EZ-Link™ Sulfo-NHS-LC-Biotinylation Kit (Thermo Scientific, Rockford, IL, USA). Biotinylated-IFN-γ was then tested using an ELISA assay. The microtiter plates were coated with 50 μL of biotinylated IFN-γ diluted in coating buffer (1 M NaHCO_3_, pH 9.6) and retained overnight at 4 °C in a moisture chamber. The coated wells were washed three times with washing buffer and blocked with blocking solution at room temperature for 1 h. After washing, 50 µL of streptavidin-conjugated peroxidase was added to each well for 1 h. The wells were then washed, and 50 µL of TMB substrate was added to each well, after which the OD was measured at 450 nm.

### 2.8. Preparation of Colloidal Gold Nanoparticles

Heat colloidal gold was synthesized using Turkevich’s method [27]. Initially, 5 mL of 5 mM chloroauric acid (Sigma-Aldrich, St.Louis, MO, USA) was added to 90 mL of deionized water. This solution was heated to boiling and vigorously stirred in a 250 mL round-bottom flask, followed by quick addition of 5 mL of 0.5% (*w*/*v*) sodium citrate (Sigma-Aldrich, St.Louis, MO, USA) to the flask. Heating continued steadily until the color of the solution turned from light yellow to black and finally red-purple. The mixture was cooled to room temperature. To adjust the pH of the colloidal gold nanoparticles, 0.2 M potassium carbonate (Univar, New South Wales, Australia) was used.

### 2.9. Synthesis of mSA-Colloidal Gold Conjugates

Gold nanoparticle-protein conjugates with varying concentrations of recombinant ^6HIS^MBP-mSA protein were prepared by following the gold nanoparticle conjugation protocol. To conjugate ^6HIS^MBP-mSA with colloidal gold, a suitable amount of protein is required to stabilize the colloidal gold particles. The colloidal gold solution was pipetted into a series of tubes, and a ^6HIS^MBP-mSA solution was added to the colloidal gold. The mixture was incubated at room temperature for 30 min. Next, 20 µL of 10% NaCl was added to each tube, and 5% (*w*/*v*) bovine serum albumin (BSA) (Sigma-Aldrich, St.Louis, MO, USA) was added to another tube in 5 mM NaCl solution. Finally, the absorbance values of the solutions were measured using UV-vis spectroscopy (at wavelengths of 520 and 580 nm), and the data were plotted.

To generate ^6HIS^MBP-mSA-colloidal gold conjugates (^6HIS^MBP-mSA-GCG), an appropriate concentration of ^6HIS^MBP-mSA [to obtain a final concentration of 500 µg/mL diluted in 5 mM sodium borate buffer (pH 9.0)] was added dropwise to the pH-adjusted colloidal gold solution (pH 7.2–7.4). The solution was then incubated for 30 min at room temperature, prior to saturating the unoccupied gold surfaces with 5 mM NaCl containing 5% BSA (*w*/*v*). The reaction was incubated at room temperature for another 30 min and then centrifuged at 4 °C to remove the unconjugated protein. The pellet was resuspended in BSA solution in 49 mM Na_2_HPO_4_ and stored at 4 °C for further experiments. Gold conjugation was confirmed by UV-vis spectrophotometry (Thermo Scientific, Waltham, MA, USA).

### 2.10. Binding Activity of the Conjugation Material

A direct dot blot immunoassay was performed to determine the binding activity of ^6HIS^MBP-mSA post conjugation. The nitrocellulose membrane was dotted with 50 µL of biotinylated IFN-γ (at concentrations of 10 and 100 µg/mL), anti-His_6_-tagged antibody (at a concentration of 10 µg/mL), and PBS. Nonspecific binding was blocked by incubating the strips in 5% (*w*/*v*) BSA in PBS. The ^6HIS^MBP-mSA-CGC solution was added to the membrane and incubated on a shaker at room temperature. The result in the positive control area was observed upon the addition of ^6HIS^MBP-mSA-CGC.

To test the complex of ^6HIS^MBP-mSA-CGC/biotinylated-IFN-γ, anti-IFN-γ mAb (clone B27), PBS, or anti-His_6_-tagged antibody was applied to a nitrocellulose membrane. The membranes were blocked with PBS containing 5% (*w*/*v*) BSA. After washing, the complex of ^6HIS^MBP-mSA-CGC/biotinylated-IFN-γ (test) or ^6HIS^MBP-mSA-CGC (control) was added to individual strips for 1 h at room temperature. Positive dots were generated upon probing with the complex of ^6HIS^MBP-mSA-CGC/biotinylated IFN-γ.

### 2.11. Evaluation of Indirect IC Strip for Anti-IFN-γ mAb Detection

To mimic the IC strip platform for human anti-IFN-γ antibody detection, a goat anti-mouse Igs-IC strip was fabricated. The capture test line was constructed by spraying goat anti-mouse Igs onto a nitrocellulose membrane at the test line zone. The ^6HIS^MBP-mSA-CGC/biotinylated-IFN-γ complex was impregnated into the conjugate pad. The immobilized nitrocellulose membrane, absorbent pad, conjugate pad, and sample pad were assembled as IC strips. The fabricated strip was dipped into anti-IFN-γ mAb (clone B27) solution, at concentrations ranging from 10–100,000 ng/mL, or irrelevant monoclonal antibody to HIV matrix protein (MA) clone G53 (anti-MA mAb, clone G53), at concentrations of 10,000 and 100,000 ng/mL. After 10 min, the presence of anti-IFN-γ mAb (clone B27) was visualized using a color tracer of colloidal gold. A positive line represented the interaction of anti-IFN-γ mAb (clone B27) with the complex of ^6HIS^MBP-mSA-CGC/biotinylated-IFN-γ at the conjugate pad. The immune complex traveled to the goat anti-mouse Igs immobilized on a nitrocellulose membrane.

### 2.12. Preparation of the IC Strip Test

The principle of the IC strip test for detection of autoAbs to IFN-γ in AOID sera was designed based on a conventional indirect ELISA (Figure 1). The ^6HIS^MBP-mSA was conjugated with colloidal gold (^6HIS^MBP-mSA-CGC) and captured with biotinylated IFN-γ. The ^6HIS^MBP-mSA-CGC/biotinylated-IFN-γ complex was coated onto a conjugate pad. Goat anti-human Igs and anti-His_6_-tagged antibodies were immobilized on the test line area and control line area, respectively. For testing, serum samples were diluted in a blocking solution. Diluted serum samples (200 µL) were dipped into each tube and allowed to travel to the absorbent pad. If the sample contained autoAbs to IFN-γ, the antibody bound to IFN-γ in the complex of ^6HIS^MBP-mSA-CGC/biotinylated-IFN-γ, which is present in the conjugate pad of the IC strip and the complex passed through the membrane. The autoAbs were captured using goat anti-human Igs at the test line position. In the meantime, the unreacted conjugates moved further on the membrane and bound to the anti-His_6_-tagged antibody on the control line, thus developing a red-purple color. Two red-purple lines, at both the test and control lines, indicated a positive result. However, in the absence of autoAbs to IFN-γ in the serum samples, the results of the strip test showed only one band at the control line. Interpretation of the test results was completed 10 min post sample application.

### 2.13. Statistical Analysis

Statistical analysis, including calculation of sensitivity and specificity of the IC strip test, was carried out using two-by-two tables [28,29].

## 3. Results

### 3.1. Validation of Recombinant IFN-γ Expression

Recombinant IFN-γ was successfully produced in the *E. coli* strain BL21 (DE3). HisTrap^TM^-purified IFN-γ was further verified using SDS-PAGE. Post purification, a major protein band was observed at a molecular weight (MW) of ~17 kDa (Figure 2A). Consequently, the crude protein and purified IFN-γ were analyzed in parallel using western blot analysis. Both crude and purified IFN-γ proteins reacted with anti-His_6_-tagged antibody and anti-IFN-γ mAb (clone B27) at the expected MW of IFN-γ (~17 kDa; Figure 2B, lane 1 and 2, respectively).

Purified IFN-γ was conjugated with biotin, which was further confirmed using ELISA. The biotinylated IFN-γ-coated wells displayed a positive signal upon probing with streptavidin-conjugated peroxidase. This result suggested that IFN-γ was successfully linked to the biotin molecules. Biotinylated IFN-γ was subsequently employed at the conjugate pad of the IC strip test.

### 3.2. Determination of ^6HIS^MBP-mSA Expression

The purified recombinant ^6HIS^MBP-mSA protein from *E. coli* BL21 (DE3) cells was verified using western blot analysis. A single band of ^6HIS^MBP-mSA at an MW of 60 kDa interacting with an anti-His_6_-tagged antibody is shown in Figure 3.

### 3.3. Detection of AutoAbs against IFN-γ Using Conventional Indirect ELISA

The levels of anti-IFN-γ autoAbs in three classified groups of samples: active AOID, inactive AOID, and normal subjects were examined using a conventional indirect ELISA. The results showed that the levels of anti-IFN-γ autoAbs present in AOID sera were diverse. The titers of anti-IFN-γ in individual patient sera were 500, 2500, and 12,500 and did not correlate with the clinical status of the patients. There was no detectable signal in the sera samples from healthy control subjects.

### 3.4. Validation of mSA-CGC/Biotinylated-IFN-γ Complex

After protein titration, an optimal concentration of ^6HIS^MBP-mSA was used in the colloidal gold conjugation process and verified using UV-vis spectrophotometry and dot blot immunoassay. The occupation of protein on the gold nanoparticles was analyzed using UV-Vis spectroscopy (Figure 4). The surface plasmon resonance properties of the AuNPs showed a peak at ~520 nm (Curve a). Whereas, the success of ^6HIS^MBP-mSA-CGC formation was demonstrated by red shift of 5 nm (Curve b). The conjugation efficiency of ^6HIS^MBP-mSA-CGC was demonstrated by the presence of red-purple dots at the biotinylated-IFN-γ and anti-His_6_-tagged antibody positions, but not at the PBS-dotted control position (Figure 5A). The intensity of the red-purple dots correlated with the amount of dotted biotinylated IFN-γ. The appearance of a red-purple dot at the anti-His_6_-tagged antibody position confirmed the successful conjugation process of ^6HIS^MBP-mSA.

In addition to proving the conjugation of ^6HIS^MBP-mSA-CGC, the interaction of the ^6HIS^MBP-mSA-CGC/biotinylated-IFN-γ complex with anti-IFN-γ mAb (clone B27) was also validated using dot blot immunoassay. After probing the ^6HIS^MBP-mSA-CGC/biotinylated-IFN-γ solution, red-purple dots appeared only at the position coated with anti-IFN-γ mAb (clone B27) and anti-His_6_-tagged antibody (Figure 5B). In contrast, the free ^6HIS^MBP-mSA-CGC solution is negatively bound to the anti-IFN-γ mAb (clone B27) (Figure 5C). These data suggested the successful establishment of the ^6HIS^MBP-mSA-CGC/biotinylated-IFN-γ complex, which was subsequently used as a reporter in the IC strip.

### 3.5. Verification of the Fabricated IC Strip Platform

To mimic the IC strip platform for human anti-IFN-γ antibody detection, goat anti-mouse Igs were immobilized on the test line. The fabricated IC strip was verified using anti-IFN-γ mAb (clone B27) spiked in PBS and human serum at various concentrations (100,000, 10,000, 1000, 100, and 10 ng/mL). As shown in Figure 6A,B, a red-purple line appeared at the test line in the presence of anti-IFN-γ mAb (clone B27) at all the tested concentrations of anti-IFN-γ mAb (clone B27) spiked PBS, while no red-purple line was observed with anti-IFN-γ mAb (clone B27) spiked human serum at 10 ng/mL. The control system using irrelevant monoclonal antibodies to MA clone G53 (anti-MA mAb, clone G53). No red-purple line was observed with anti-MA mAb (clone G53)-spiked at any of the concentrations.

### 3.6. Evaluation of Clinical Samples Using IC Strip Test

The fabricated IC strip test was validated by comparing its results for 66 serum samples with those obtained using conventional ELISA. Among these, 36 samples were from AOID patients, while the rest were normal healthy subjects (titer < 500). The results showed that the intensity of the test line corresponded with the anti-IFN-γ autoAb titer determined using conventional ELISA, as shown in Figure 7. There was no positive band on the test line in the case of all the normal healthy subjects. However, there was no relationship between antibody level and the status of the patients.

When the cut-off was set at a conventional ELISA titer of 2500, the sensitivity and specificity of the IC strip test were 84% and 90.24%, respectively, with a positive predictive value (PPV) of 84.00 and a negative predictive value (NPV) of 90.24 (Table 1). When a cut-off titer of 500 was selected, the sensitivity and specificity were 73.52% and 100%, respectively, with PPV of 100 and NPV of 78.04 (Table 2).

## 4. Discussion

Anti-cytokine autoAbs induce pathological symptoms in defective immune systems [4,5,30]. Therefore, it is crucial that anti-cytokine autoAbs be precisely detected. In 1998, an ELISA was developed to identify anti-IFN-γ autoAbs [31]. Since then, an indirect ELISA is generally used for the detection of anti-IFN-γ autoAbs, to forecast the clinical status of patients [11,14]. Currently, various sophisticated technologies including multiplex cytokine assays have been developed for the detection of anti-cytokine autoAbs samples such as anti-Granulocyte-Colony Stimulating Factor, anti-IFN-β, and anti-IFN-γ [32]. However, these methods are not practical for screening purposes.

Recently, Rattanathammethee et al. developed a dot ELISA to detect anti-IFN-γ Abs in AOID patients [17]. Although the ELISA washer and reader steps were omitted in this assay, it still required a multi-step process. Therefore, the one-step assay developed in the present study, which is a rapid throughput analysis, should be considered. Herein, an IC strip has been used for detecting anti-IFN-γ autoAbs and providing a potential early diagnosis for AOID patients. Initially, the standard method was used to conjugate IFN-γ onto colloidal gold nanoparticles. However, the direct conjugation was unsuccessful. Some reports have described the parameters that affect the efficiency of protein-bound colloidal gold, such as the amount of protein, buffer type, pH of gold, and antibody. Generally, the optimal pH is close to the isoelectric point (pI) of the protein to be conjugated [33] or slightly higher than the pI of the labeled material [26]. The unsuccessful colloidal gold labeling of IFN-γ could be attributed to the pI of the IFN-γ protein (~8.7) [34] being higher than the pH of colloidal gold used (7.2), or the composition of the amino acids of IFN-γ. Thus, biotinylated recombinant IFN-γ was generated for further indirect labeling with colloidal gold-conjugated streptavidin.

Streptavidin-biotin or anti-biotin antibody-biotin are generally assembled in lateral flow assays, to increase the number of bound colloidal gold labels [19,35,36]. However, this strategy is not cost-effective to achieve high yield production of biologically active streptavidin. Several downstream processes for protein denaturation and protein folding are needed in this case [37,38,39]. Recently, the fusion protein ^6HIS^MBP-mSA has been produced in *E. coli*. Owing to its specific biotin interaction, similar to that observed in the case of native streptavidin, it has been formerly applied in flow cytometry analysis and ELISA assay [21,22]. Therefore, in the present study, ^6HIS^MBP-mSA protein was subjected to indirect conjugation with biotinylated recombinant IFN-γ and used as a reporter for IFN-γ autoAb detection in the IC strip. Theoretically, one colloidal gold particle is occupied by hundreds of protein molecules, that is, IgGs [40]. In contrast, our indirect conjugation strategy provides an advantage by allowing for deposition of more than one colloidal gold particle per mole of IFN-γ. Not only do individual biotin molecules on IFN-γ conjugate with ^6HIS^MBP-mSA-CGC, but ^6HIS^MBP-mSA-CGC also promotes lattice formation of biotinylated IFN-γ (Figure 5). This conjugated complex should enhance the sensitivity of the assay, as previously reported [41,42].

After conjugation, ^6HIS^MBP-mSA-GCG was examined using UV-vis spectrophotometry and dot blot immunoassay. The spectrum shift data in Figure 4 confirmed the presence of protein on the colloidal gold surface. The direct dot blot immunoassay confirmed that ^6HIS^MBP-mSA-CGC retained its biological activity in reacting specifically with biotinylated-IFN-γ (Figure 5A). In addition, the interaction of the ^6HIS^MBP-mSA-CGC/biotinylated-IFN-γ complex and anti-IFN-γ mAb was observed (Figure 5B,C). The analytical sensitivity of the IC strip was further determined using various concentrations of anti-IFN-γ mAb (clone B27), ranging from10 to 100,000 ng/mL. The red-purple band immobilized with the goat anti-mouse antibody gradually reduced in its intensity but still remained visible until 10 ng/mL of anti-IFN-γ mAb spiked in PBS (Figure 6A), whereas the sensitivity decreased to 100 ng/mL when anti-IFN-γ mAb was spiked in human serum (Figure 6B). In addition, it implied that anti-IFN-γ mAb (clone B27) specifically captured ^6HIS^MBP-mSA-CGC/biotinylated-IFN-γ, without falsification from the irrelevant mAb. The IC strip fabricated in the present study displayed a comparative detection sensitivity to that reported for the commercialized human anti-IFN-γ ELISA kit, which is a detection range of 3.12 ng/mL-200 ng/mL [43]. Consequently, an IC strip was fabricated for the detection of anti-IFN-γ autoAbs in AOID patients. The performance of the IC strip for detecting anti-IFN-γ autoAbs in AOID sera was evaluated and compared to that obtained using conventional ELISA (Figure 7). Noticeably, the level of IFN-γ autoAbs from 36 individual AOID serum samples in this cohort did not determine the pathologic condition of individual patients. Nevertheless, the results obtained using the IC strip were consistent with those obtained using conventional indirect ELISA; in addition, the IC strip did not detect any false positives in the serum samples from normal subjects.

The intrinsic advantage of ELISA is disease monitoring and prognosis based on antibody levels. It has been reported that the level of anti-IFN-γ antibodies in inactive AOID patients is noticeably lower than that in active AOID patients [5]. Moreover, the level of autoAbs to IFN-γ has been considered as a biomarker for the severity prediction of disseminated NTM disease [15,44] and drug response for *Mycobacterium abscessus* infection [45]. Relying on the cut-off of a conventional ELISA titer, i.e., 2500, the sensitivity and specificity of the IC strip test were found to be 84% and 90.24%, respectively. The PPV and NPV were 84.00 and 90.24, respectively. The specificity reached 100%; however, the sensitivity markedly reduced to 73.52% with a PPV of 100 and NPV of 78.04 when a cut-off titer of 500 was used. To date, no suggested cut-off has been proposed. Thus, the IC strip should be validated in conjunction with ELISA against a cohort, with a detailed clinical time course to adjust its efficacy. If a suitable ELISA cut-off titer is defined, it will support the formulation of IC strip components to acquire the appropriate performance.

Most anti-IFN-γ autoAbs cases have been reported from university hospitals since laboratory diagnosis requires special instruments and well-trained staff. The actual number of AOID cases may be substantially higher than expected, owing to the association of certain HLA phenotypes [46]. The user-friendly, rapid, and robust IC strip developed in the present study will support better clinical management for this high fatal outcome syndrome. However, several studies suggest that detection of the presence and level of anti-IFN-γ autoAbs is not sufficient for clinical manifestation, because autoAbs with neutralizing capacity are required for pathogenicity [15,47]. Therefore, a simplified assay to confirm the presence of functional activities of the anti-IFN-γ autoAbs also needs to be explored in the future.

## 5. Conclusions

To overcome the inefficient direct colloidal gold conjugation of recombinant IFN-γ, passive labeling was performed using a ^6HIS^MBP-mSA partner. ^6HIS^MBP-mSA-CGC/biotinylated-IFN-γ was successfully synthesized and retained its antigenicity. This modified approach enhanced the sensitivity of the IC strip, resulting from the network of avidin-biotin interaction. As a consequence, the detection sensitivity reached the nanogram level, which is comparable to that of conventional ELISA. Owing to the high solubility of ^6HIS^MBP-mSA produced from *E. coli* cytoplasm, the functional structure was obtained without refolding steps. The ^6HIS^MBP-mSA-CGC/biotinylated-IFN-γ was impregnated with the conjugate pad as a tracer agent in the IC strip test. The fabricated IC strip was substituted for conventional indirect ELISA, to detect anti-IFN-γ autoAbs in AOID sera. The distinctive performance of the IC strip was demonstrated by relying on an arbitrary cut-off ELISA titer. Due to the lack of clinical progression information of the AOID cohort in this study, the efficiency of the IC strip needs to be further customized to a clinically significant ELISA titer. The developed IC strip provides an opportunity for early diagnosis of AOID, without the requirement of sophisticated instruments. In addition, the correlation between anti-IFN-γ autoAbs and clinical manifestations will benefit therapeutic monitoring.

## Figures and Tables

**Figure 1 diagnostics-11-00987-f001:**
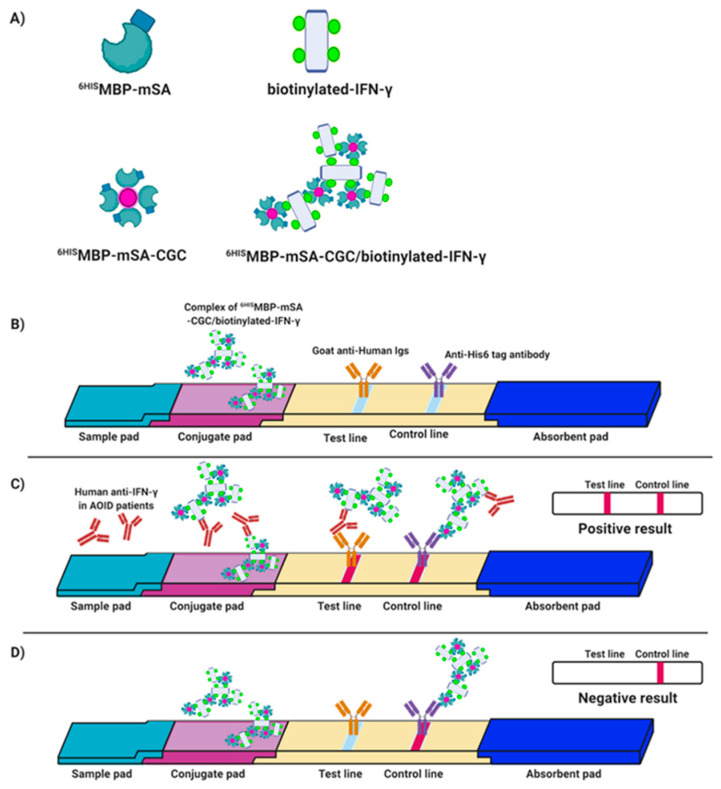
Principle of the IC strip test for detection of autoAbs to IFN-γ in AOID sera. The symbols of each component are defined i.e., ^6HIS^MBP-mSA, ^6HIS^MBP-mSA-CGC, biotinylated-IFN-γ, and ^6HIS^MBP-mSA-CGC/biotinylated-IFN-γ (**A**). Schematic diagrams of assembled components (**B**), positive result (**C**), and negative result (**D**) are depicted. The appearance of red-purple color at both the test and control lines is interpreted as a positive result. The appearance of red-purple color only at the control line is interpreted as a negative result. Of note, the illustration presented in this figure was generated by the author, using a combination of elements available from the Biorender site (https://app.biorender.com, accessed on 25 March 2021).

**Figure 2 diagnostics-11-00987-f002:**
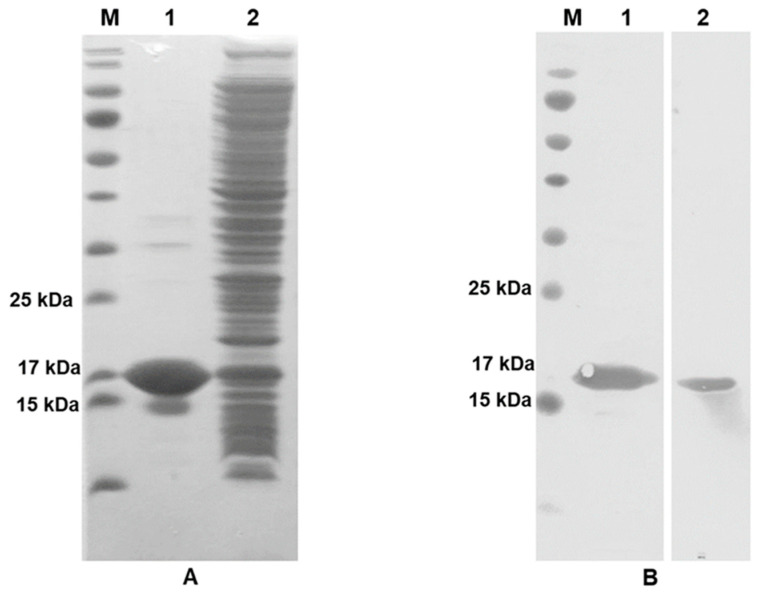
Validation of IFN-γ production in crude and purified forms; PAGEBlue™ staining (**A**), western immunoblot (**B**) using an anti-His_6_-tagged antibody (1) and anti-IFN-γ mAb (clone B27) (2). After the primary antibody reaction, the membranes were finally incubated with goat anti-mouse Igs tagged with HRP and TMB membrane substrate.

**Figure 3 diagnostics-11-00987-f003:**
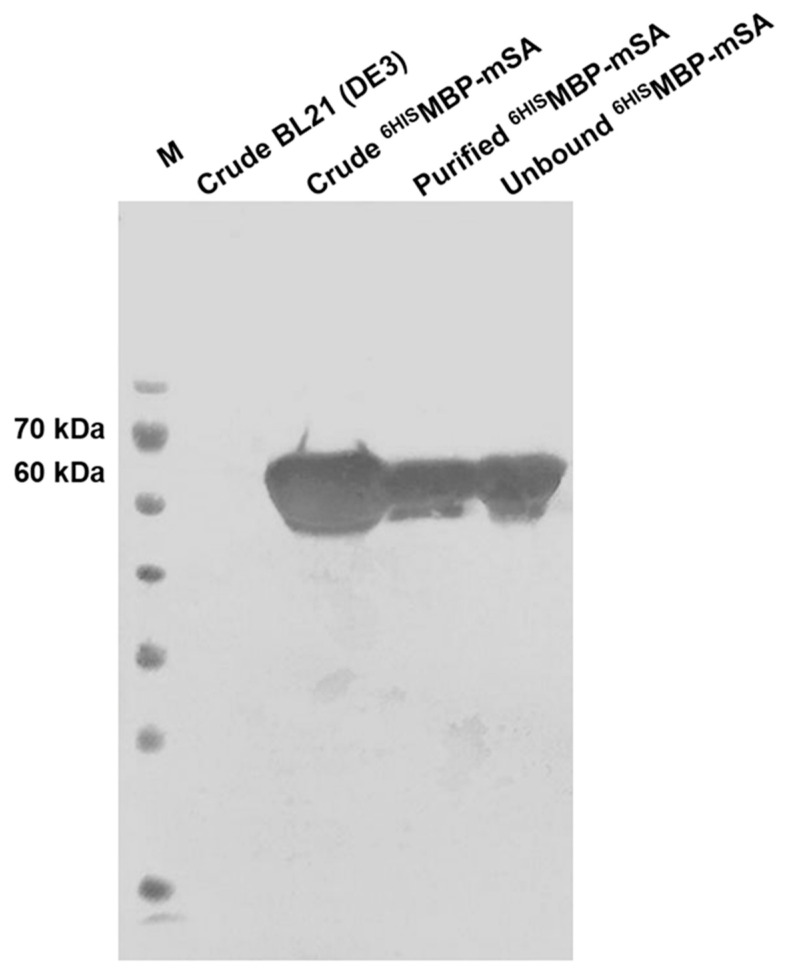
Verification of the recombinant ^6HIS^MBP-mSA protein using western immunoblot. Crude extracts from BL21 (DE3) and BL21 (DE3) cloned with pET-MBP-mSA2 plasmid and purified ^6HIS^MBP-mSA were probed with anti-His_6_-tagged antibody followed by goat anti-mouse Igs conjugated with HRP. The positive signal was generated using TMB membrane substrate.

**Figure 4 diagnostics-11-00987-f004:**
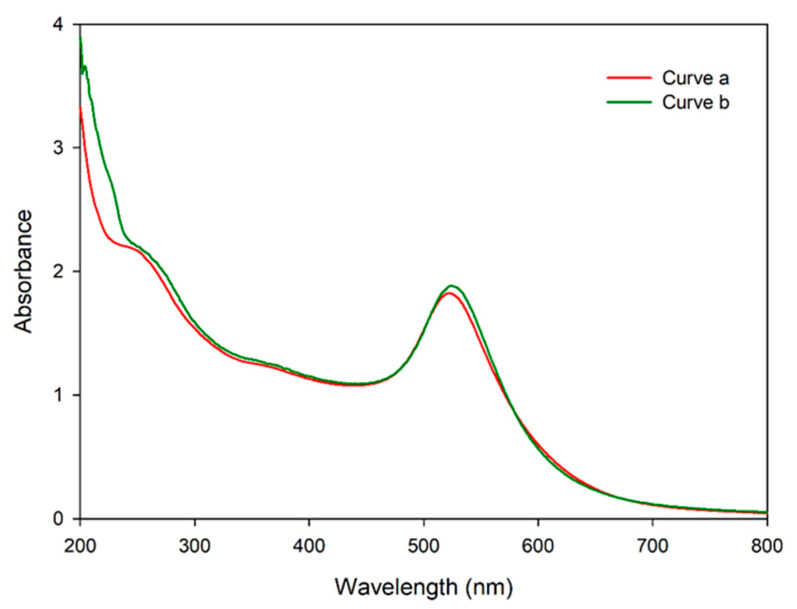
The characterization of colloidal gold before and after ^6HIS^MBP-mSA proteins adding using UV–VIS spectroscopy. Curve a, colloidal gold; curve b, ^6HIS^MBP-mSA-CGC. UV-vis spectroscopy confirmed the conjugation as showed the red shift in the peak absorbance wavelength on conjugation.

**Figure 5 diagnostics-11-00987-f005:**
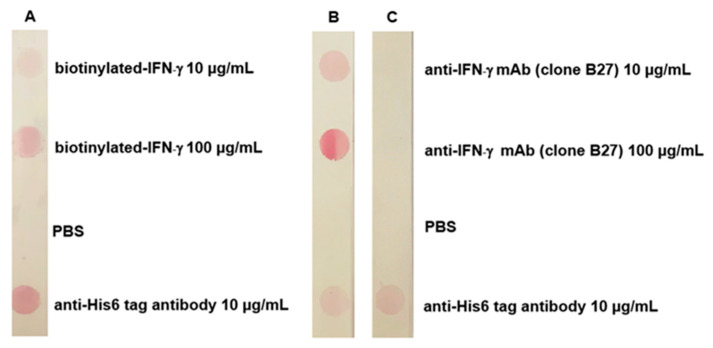
Assessment of ^6HIS^MBP-mSA-CGC and ^6HIS^MBP-mSA-CGC/biotinylated-IFN-γ using dot blot immunoassay. The binding activity of ^6HIS^MBP-mSA-CGC was analyzed by means of direct interaction with dotted biotinylated-IFN-γ, anti-His_6_-tagged antibody, and PBS (**A**). In addition, the complex of ^6HIS^MBP-mSA-CGC/biotinylated-IFN-γ was further evaluated against dotted anti-IFN-γ mAb (clone B27), anti-His_6_-tagged antibody, and PBS (**B**). The control system was ^6HIS^MBP-mSA-CGC (**C**).

**Figure 6 diagnostics-11-00987-f006:**
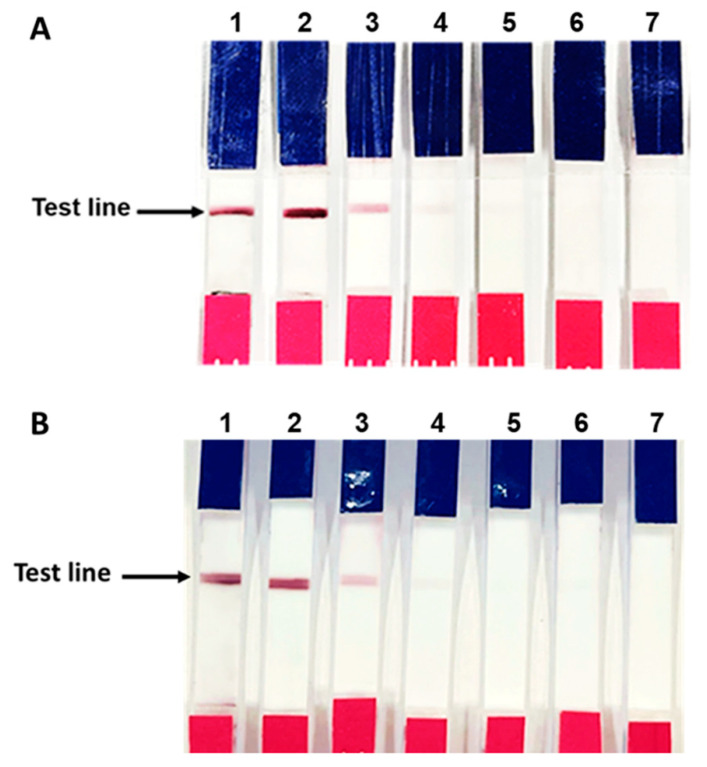
Performance of the IC strip platform. The IC strip was fabricated by immobilizing the goat anti-mouse Igs antibody on the test line. ^6HIS^MBP-mSA-CGC/biotinylated-IFN-γ was sprayed on the conjugate pad. The assembled strips were dipped into anti-IFN-γ mAb (clone B27) spiked in PBS (**A**) and human serum (**B**) at concentrations of 100,000, 10,000, 1000, 100, and 10 ng/mL (1–5), along with irrelevant anti-MA mAb (clone G53) at concentrations of 100,000 and 10,000 ng/mL (6–7).

**Figure 7 diagnostics-11-00987-f007:**
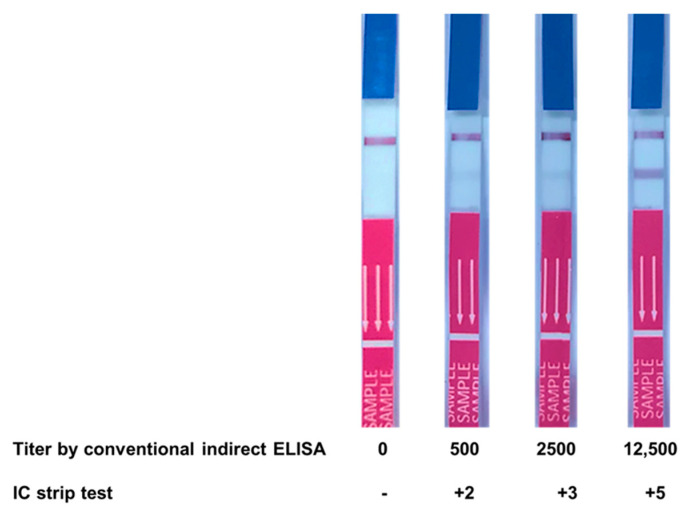
Correlation of IC strip results at different titers of conventional ELISA. The IC strip test was used to detect anti-IFN-γ autoAbs in patient sera with ELISA titers of 500, 2500, and 12,500 as well as in normal serum, following which the band intensity was observed.

**Table 1 diagnostics-11-00987-t001:** Lateral-flow assay efficacy comparison at a conventional ELISA cut-off titer of 2500.

IC-Strip	ELISA		Total
	Titer ≥ 2500	Titer < 2500	
Positive	21	4	25
Negative	4	37	41
Total	25	41	66

**Table 2 diagnostics-11-00987-t002:** Lateral-flow assay efficacy comparison at a conventional ELISA cut-off titer of 500.

IC-Strip	ELISA		Total
	Titer ≥ 500	Titer < 500	
Positive	25	0	25
Negative	9	32	41
Total	34	32	66

## Data Availability

Not applicable.

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
