# Peer review of "Latticed Gold Nanoparticle Conjugation via Monomeric Streptavidin in Lateral Flow Assay for Detection of Autoantibody to Interferon-Gamma"

_diagnostics, 2021, doi:10.3390/diagnostics11060987_

Round 1

Reviewer 1 Report

In this manuscript, Thongkum et al reported a gold nanoparticle-based lateral flow assay for the detection of IFN asscociated with AOID. Although lateral flow assay has been developed for IFN detection, the incorporation of gold nanoparticle and 6HIS-maltose binding protein-monomeric streptavidin is novel and may enhance its sensitivity. Both sensitivity and selectivity of the assay were characterized, with nanogram level of sensitivity. Clinical samples were assessed using their assay as well. Here are a few minor concerns:

  1. What is the reproducibility of the gold nanoparticle-based flow assay, especially at the nanogram concentration and in the complex biofluids? I would suggest the authors to perform biological replicates and provide the standard deviation of the signals.
  2. Is there a way to quantify the signals on their flow assay? Qualitative characterization of IFN is good, but if they can prove the linearity of their assay in quantitative detection of IFN, it can provide more accurate information to patients and doctors.
  3. How is the selectivity at lower concentrations? The selectivity seems to be ok according to Figure 6 but it would be better to quantify.
  4. In Table 2, is it Titer>=2500, or should it be Titer>=500?
  5. What is the direct proof of the successful conjugation of MBP-mSA to gold nanoparticle? Plasmonic peak shift may be used to support this. UV-Vis was mentioned in the Methods section but I did not see the data mentioned in the Results section.
  6. The sentence on Page 1, Line 32-34 is confusing and should be more clearly written. What does sensitivity of 84% mean?
  7. In the introduction, advantages of gold nanoparticles in flow assay should be introduced and cited, as it has been widely used in lateral flow assays.
  8. Validation of their assay method using clinical samples is good, but are these tests on clinical samples performed blindly? 

Reviewer 2 Report

The presented article “Latticed gold nanoparticle conjugation via monomeric streptavidin in lateral flow assay for detection of autoantibody to interferon gamma” is well-written research article. Many methods are included and interesting scheme is described. Before the publication some modifications are required.

  1. Page 3 of 15 – lines 104-106. Ethical statement should be moved to the corresponding part of Methods (or Results and Discussion).
  2. Section 2.2. The reason of addition of ampicillin to the both medium should be described (in a few words).
  3. Page 3 of 15 – lines 106, 119, 124-129, page 4 of 15 – lines 175. Please, check the brackets (other direction).
  4. Description of MSA should be added at the first mention.
  5. Section 3.2 – lines 277-278. Type of cells should be added to the materials section as well as their origin.
